# Nanoplastics and Neurodegeneration in ALS

**DOI:** 10.3390/brainsci14050471

**Published:** 2024-05-07

**Authors:** Andrew Eisen, Erik P. Pioro, Stephen A. Goutman, Matthew C. Kiernan

**Affiliations:** 1Division of Neurology, Department of Medicine, University of British Columbia, Vancouver, BC V6S 1Z3, Canada; erik.pioro@ubc.ca; 2Department of Neurology, University of Michigan, Ann Arbor, MI 48109, USA; sgoutman@med.umich.edu; 3Neuroscience Research Australia, Randwick, Sydney, NSW 2031, Australia; matthew.kiernan@neura.edu.au

**Keywords:** ALS, micro/nanoplastics, exposome, gut–brain axis, TDP-43

## Abstract

Plastic production, which exceeds one million tons per year, is of global concern. The constituent low-density polymers enable spread over large distances and micro/nano particles (MNPLs) induce organ toxicity via digestion, inhalation, and skin contact. Particles have been documented in all human tissues including breast milk. MNPLs, especially weathered particles, can breach the blood–brain barrier, inducing neurotoxicity. This has been documented in non-human species, and in human-induced pluripotent stem cell lines. Within the brain, MNPLs initiate an inflammatory response with pro-inflammatory cytokine production, oxidative stress with generation of reactive oxygen species, and mitochondrial dysfunction. Glutamate and GABA neurotransmitter dysfunction also ensues with alteration of excitatory/inhibitory balance in favor of reduced inhibition and resultant neuro-excitation. Inflammation and cortical hyperexcitability are key abnormalities involved in the pathogenic cascade of amyotrophic lateral sclerosis (ALS) and are intricately related to the mislocalization and aggregation of TDP-43, a hallmark of ALS. Water and many foods contain MNPLs and in humans, ingestion is the main form of exposure. Digestion of plastics within the gut can alter their properties, rendering them more toxic, and they cause gut microbiome dysbiosis and a dysfunctional gut–brain axis. This is recognized as a trigger and/or aggravating factor for ALS. ALS is associated with a long (years or decades) preclinical period and neonates and infants are exposed to MNPLs through breast milk, milk substitutes, and toys. This endangers a time of intense neurogenesis and establishment of neuronal circuitry, setting the stage for development of neurodegeneration in later life. MNPL neurotoxicity should be considered as a yet unrecognized risk factor for ALS and related diseases.

## 1. Introduction

Amyotrophic lateral sclerosis (ALS) is a progressive neurodegeneration causing paralysis of skeletal and respiratory muscles and cognitive decline [1,2]. ALS mechanisms are broad [3], and the molecular subtypes underlying disease vary among patients [4]. In parallel, genetic variability in ALS is also inconstant; most ALS patients lack a disease-causing monogenic mutation, but numerous risk-genes have been identified, each having small effects [5], emphasizing the need to identify non-genetic ALS risk factors. Of increasing concern is the role the life course of environmental exposures—the exposome—plays in ALS risk [6]. This is especially important as these exposures may be modifiable and offer the hope of disease prevention for ALS [7], and other neurological diseases more broadly [8]. 

Exposures to toxicants or xenobiotics negatively impact health directly, by impairing chemical reactions or enzymes, impairing ion channel function, disrupting the cellular membrane, or causing oxidative stress or inflammation [9,10,11]. Indirect health effects may result from the metabolic products of a toxicant, as opposed to the toxicant itself. Alternatively, exposures can also promote epigenetic modifications [12,13], which alter the output without changing the DNA itself. Epigenetic modifications include DNA methylation, histone modifications [14], and non-coding RNAs and microRNAs [15].

Mislocalization and extra-nuclear aggregation of TAR DNA-binding protein 43 (TDP-43) is a pathological hallmark of ALS. Exposure to environmental toxicants, including dioxins, polychlorinated biphenyls (commonly used in plastics), and polycyclic aromatic hydrocarbons or heavy metal neurotoxicants increases the level of TDP-43, providing a link between environmental factors and TDP-43-associated disorders [16]. 

Even though plastic production has increased exponentially since the 1950s [17,18], recognition of microplastics as pollutants of risk to humans is only recent [19,20], and their potential as a risk factor in the pathogenesis of neurodegeneration has received very limited attention. Given the ubiquitous availability of MNPLs in daily life, this may be an emerging ALS risk given the overlap with ALS mechanisms, and thus there is a need for further research. Human exposure to microplastics is predominantly through ingestion, which can negatively impact the gut microbiome and gut–brain axis [21,22] and alterations of gut microbiota may contribute to the etiology of ALS and its progression [23,24,25].

Here, we justify the concern of MNPLs in the setting of ALS exposome research, discuss sources of micro and nanoplastics and their toxicologic properties in humans, and the potential mechanistic links with ALS.

## 2. The Exposome

As noted by Wild [26], the “exposome encompasses life-course environmental exposures (including lifestyle factors), from the prenatal period onwards”. Wild reflected that an increased burden of human disease may result from more prevalent environmental exposures interacting with potentially frequent but low-penetrant genetic variants [27], and advocated for improved exposure biomarkers [26]. There is an increasing appreciation of the role the exposome plays in influencing neurodegenerative diseases [8], along with a greater emphasis on research supporting the neural exposome [28]. In ALS, there are recurrent exposure types that contribute to disease risk, [6] such as pesticides, heavy metal exposure, and physical activity. A complex aspect of this exposure research is accounting for the combined effects of multiple exposures on a health outcome, such as via the use of environmental risk scores [29,30,31]. Certainly, ongoing research is needed to identify specific exposure that most influences risk either alone or in combination with other related or disparate exposures [32].

## 3. Micro- and Nanoplastics (MNPLs)

The term microplastics was coined in 2004 and used to describe small plastic particles. However, there is still no all-inclusive definition that accurately encompasses all criteria that could potentially describe what a microplastic is [33]. Microplastics are defined as “…any synthetic solid particle or polymeric matrix, with regular or irregular shape, and with size ranging from 1 μm to 5 mm, of either primary or secondary manufacturing origin, which are insoluble in water” [33]. Plastics are of many different shapes, sizes, and colors, made from polymers with multiple chemical additives [34]. Primary particles are released directly into the environment from many sources, whereas secondary (weathered) particles result from degradation and fragmentation [35,36]. Opening plastic packages can generate microplastics in daily life, regardless of the method of opening and plastic target [37]. Most plastic polymers are low-density polyethylene, polyvinyl chloride, polystyrene, polypropylene, and polyethylene terephthalate [9,38]. Non-stable plastics are subject to fragmentation through photodegradation and erosion, forming toxic micro/nano plastics [39]. Not only do plastics contain hazardous chemicals through their manufacture, including plasticizers (e.g., bisphenol-A), UV stabilizers, lubricants, dyes (e.g., heavy metals), and flame retardants, they also serve as vectors for some of these same toxic chemicals found in the environment [40,41].

Biodegradable plastics have been commercialized in the manufacturing of various types of products such as garbage bags, compost bags, poly bags and agricultural mulch films and can decompose after disposal to the environment by biodegradation processes involving the use of enzymes produced by bacteria and microorganisms to break the plastic chemical bonds, producing CO_2_ and H_2_O. This leads to large numbers of plastic particles with greater surface areas for interactions with the surrounding environment [42]. However, of greatest concern are those synthesized from petroleum [43]. Common plastic polymers found in the environment include polyethylene, polypropylene, polyvinyl chloride, and polyethylene terephthalate [44]. Although plastics are highly durable, they can degrade over time, releasing microplastics (1 µm–5 mm particles) and nanoplastics (<1 µm particles) [43]. 

## 4. Sources of MNPLs

MNPLs accumulate in the environment and are detectable in air, water, and soil [43,44,45,46]. Human exposure routes are dermal, oral ingestion (contaminated water, food, or dust or release from plastic packaging), and inhalational [42,44,47]. Ingestion is responsible for the greatest exposure in humans [47]. Of concern is that the levels of MNPLs in the environment are projected to increase [48].

Small oceanic plastic fragments were first reported in the 1970s when Thor Heyerdahl, the Norwegian adventurer and ethnologist, captained an expedition across the Atlantic Ocean from Morocco to Barbados. Over the course of his journey, he encountered abundant plastic waste [49]. Microplastics are widespread in all ecosystems [34,50,51,52] including in many human foods and drinking water [53] and are defining a new epoch dubbed the Plasticene [54]. When airborne, they can travel long distances from their original source [55]. MNPLs can be purposely produced (primary), as a consequence of fragmentation and degradation (secondary) or released from synthetics microfibers of clothing [43]. Primary sources include industrial detergents and cosmetics [44] (see Table 1). Further, plasticizers including phthalic acid esters (PAEs) can co-pollute, thereby increasing the toxicity of these particles [44]. MNPLs in soil and water can also absorb pathogenic heavy metals [56], and organic pollutants, such as persistent organic pollutants [44,45,57].

## 5. Human Exposure

Humans are primarily exposed to MNPLs through ingestion, and estimates indicate that humans ingest tens of thousands to millions of MNPLs annually (several milligrams daily). The main plastic-containing ingestants are drinking water, seafood, honey, beer, table salt, and milk [58], and recently identified leafy vegetables. Inhalation [59,60] and dermal contact [61] occur less frequently [58,62,63,64], and some developed drug delivery systems also expose human beings to MNPLs via parenteral routes, including intravenous and intracranial/brain application [65]. Neonates and infants are exposed to MNPLs through breast milk and milk substitutes [66]. Plastic bottle caps, plastic teabags and infant feeding bottles can release a considerable number of MNPLs [62,67] (see Figure 1).

MNPLs have been isolated from many human tissues, including stool [68], blood (even after a single oral exposure [69]), lung [70,71], breast milk and formula [66], and colon [72,73]. Contaminated vascular tissues allow microplastic transportation to human tissues [74], and evidence confirms that ultrafine plastics cross the blood–brain barrier (BBB) [75]. During exposure, non-plastic particles acquire an environmental eco-corona consisting of biomolecules, organic matter, and chemical and biological contaminants [52]. These accumulate on the surface of plastic particles when they are exposed to biological fluids [76,77] and the protein–plastic interaction enables passage through the BBB [78]. Alternatively, MNPLs may penetrate the brain directly through the nasal olfactory pathway. This route has been shown to cause neurotoxicity in animal models such as marine invertebrates, fish, and rodents [9,79]. Some small-sized polystyrene nanoplastics can enter neurons by endocytosis and accumulate in the cytoplasm [80].

## 6. Neuroinflammation and MNPLs

Chronic inflammation is a key feature in the pathogenesis of ALS, and other neurodegenerative diseases [81,82], and is the principal toxic response to MNPLs in all tissues studied [83,84,85]. After absorption, their interaction with neurons and glia is dependent on their surface properties and the biological molecules they encounter, including carbohydrates, proteins, and phospholipids, and they form a ‘crown’ called a protein corona [20]. In vitro studies indicate that polystyrene nanoparticles coated with a protein corona facilitate translocation and may alter its form and characteristics, potentially increasing cell interactions and toxicity [86].

Due to their post-mitotic nature, and inability to regenerate, neurons may be more vulnerable to cellular toxicity and are more ATP-dependent than other cells, making them more susceptible to energy crises associated with inflammation [87]. The neuroinflammatory cascade is activated by microglia and astrocytes and mediated by key pro-inflammatory cytokines (IL-1β, IL-6, and TNFα), which regulate adhesion molecule expression, cell growth, cell division, and apoptosis [88]. The response is further mediated by extracellular signaling chemokines (CCL2, CCL5, CXCL1), secondary messengers (nitric oxide and prostaglandins), and reactive oxygen species (ROS) [89].

Despite extensive recognition of plastic particles in most human tissues, studies exploring the inflammatory response to toxicity have largely relied on assumptions from animal models or in vitro cell culture models [9,20,40]. In vitro experiments with a human brain-derived microglial cell line (HMC-3) exposed to secondary (weathered) MNPLs demonstrated a severe inflammatory response [90], but the extent of neurotoxic outcomes has been limited by lack of comparison to different particle types, shapes, sizes, and exposure concentrations [51,83]. In aquatic species, MNPL neurotoxicity induces a classic inflammatory response with increased pro-inflammatory cytokine production, oxidative stress with generation of ROS, and mitochondrial dysfunction [91]. Neurotransmitter systems, including glutamate, GABA serotonin, histamine, and ATP are also impaired [10,92], and these are known to mediate pathophysiological functions of microglia in ALS [93].

## 7. MNPL Neonatal Toxicity

The abnormal biological cascades that culminate in ALS predate the clinical disease by years or decades [94], and it has been hypothesized that the seeds for sporadic ALS may evolve as early as the neonatal period [95]. The neonatal and perinatal periods engender complex neuronal activity, intense neurogenesis, establishment of neuronal circuitry, and cell migration with neuronal differentiation and sprouting of axonal connections [95]. Even small doses of MNPLs can impact these processes, setting the stage for neurodegenerative disorders in later life [64,96]. The corticomotoneuronal system, key in the pathogenesis of ALS [97], because of its large Betz multi-synaptic neurons, is particularly vulnerable to toxicity during development [98]. 

Neonates are continuously exposed to MNPLs through everyday items such as breast milk, cow milk, and infant milk powder, as well as plastic-based products like feeding bottles and breast milk storage bags [99]. Studies of human stool, fetus, and placenta provide direct evidence of exposure to MNPLs in infants and children [100], and they are detectable in breast and human milk substitutes [66]. MNPLs have been identified in amniotic fluid during pregnancy [101]. A recent high-resolution ex vivo MRI study in mice revealed brain abnormalities, including in the motor cortex and corpus callosum (both predominantly involved in ALS), following maternal exposure to polystyrene nanoplastics [102]. Overall there is mounting evidence that exposure to MNPLs is detrimental during the perinatal period, particularly through ingestion, and can cause toxicity specifically in primary motor neurons by activating an oxidative stress response and inducing apoptosis, consequently impairing motor performance [103].

During embryonic and neonatal periods, epigenetic mechanisms are responsible for shaping early developmental programming of the central nervous system, which is particularly sensitive to a toxic environment. Epigenetics maintains the memory of a phenotype chromosome without alterations in the DNA sequence, bridging environmental stimuli and gene expression and environmental exposure to toxins such as nanoplastics. Epigenetic modification fine tunes gene expression in response to changes in the environment. The alterations result in chromatin remodeling, direct covalent modifications to the DNA itself, post-translational modifications to histone proteins, and activity of noncoding RNAs [104]. Normally, DNA and histone modifications are erased and re-established in each generation through a developmental reprogramming, and lifetime epigenetic changes are not generally inherited by subsequent generations. However, there is evidence that at least some epigenetic alterations can escape re-establishment of the epigenome with alterations in the epigenetic profile occurring in the offspring of exposed individuals indicative of intergenerational inheritance [105,106]. Further investigation is required to determine the specificity of MNPLs in relation to epigenic modifications.

## 8. The Gut–Brain Axis and MNPLs

In humans, oral ingestion is the prime means of exposure to MNPLs, and the gastrointestinal tract constantly interacts with these small particles [107]. Micro/nano plastics induce inflammation in the gastrointestinal system, promoting dysbiosis of the gut microbiome [83,84]. An abnormal gut microbiome and the gut–brain axis have been implicated in the pathogenesis of all major neurodegenerative disorders (Alzheimer’s disease, Parkinson’s disease, Huntington’s disease, and ALS) [108,109]. Gut dysbiosis in ALS was initially reported in 2015 [110], and over the past decade, further reports have described an impaired gut microbiome in ALS patients and rodent models [23,24,25,111,112,113].

The gut–brain axis is a bidirectional network [114,115]. Communications between the gut microbiome and the central nervous system are multifaceted involving the vagus nerve, which transmits neural signals from the brain to the gut, vasculature, immunological, lymphatic and glymphatic systems, and the hypothalamic–pituitary–adrenal axis. The gut microbes and their metabolites and the gut neurotransmitters and hormones secreted by enteroendocrine cells can be transported to the brain, interacting with the host immune system. They include glutamate, dopamine, and acetylcholine, and inhibitory γ-aminobutyric acid (GABA). There are also intermediate compounds, notably short chain fatty acids and tryptophan [116]. Signals generated by these neurotransmitters and molecules are transported to the brain via vagus nerve afferents [114,117]. In response, the brain signals back to enterochromaffin cells and enteroendocrine cells in the gut wall, and the mucosal immune system via vagus efferents (See Figure 2). 

Gut microbe-derived metabolites traverse the gut barrier and activate innate immune cells, with an increase of proinflammatory cytokines (TNF-α, IL-1β, IL-6, amongst others) inducing subsequent neuroinflammation. The same metabolites can traverse an impaired blood–brain barrier and interact with microglia in the brain, exacerbating neuroinflammation [118]. 

MNPLs can directly break through the gut barrier. The inner mucus layer of the gut acts as a barrier protecting the underlying epithelial cells from toxicants including MNPLs. When they reach the inner mucus layer, they induce development of biofilms with complex bacterial communities, which subsequently degrade its integrity and penetrate the epithelial barrier [119]. There is limited human information regarding this, but animal studies confirm that exposure to MNPLs causes oxidative damage and inflammation and immune cell toxicity in the gut, with destruction of the gut epithelium [120]. As a result, MNPLs can translocate to secondary structures, including the brain [121]. Several species develop nanoplastic-induced gut dysbiosis with health impacts [21]. This occurs in zebrafish [122,123,124], crayfish [125], and mice [126,127,128], and exposure to polyethylene microplastics also affects the immature human and non-human gut microbiome [129,130]. The gut microbiome influences neurodevelopment and when impaired can cause neurodevelopmental disorders [131,132].

As discussed further below, truncation, abnormal aggregation and mislocalization to the cytoplasm of TDP-43 encoded by the *TARDBP* gene, an RNA-binding protein that predominantly localizes to the nucleus, are hallmarks of ALS pathology [133]. Aggregates of TDP-43 are considered causative of ALS and frontotemporal dementia. When TDP-43 becomes mutated or mislocalized out of the nucleus of neurons and glial cells and forms cytoplasmic inclusions, it can lead to RNA splicing dysregulation. This in turn can result in the generation of an altered transcriptome and proteome within the neuron, changing the diversity and quantity of gene products. In the early stages of ALS, soluble cytoplasmic TDP-43 is found in the large pyramidal neurons, including Betz cells within the motor cortex [98]. This toxic TDP-43 subsequently spreads to bulbar and spinal motoneurons with TDP-43 aggregate formation [134]. 

Recent evidence indicates TDP-43 aggregates occur in several non-central nervous tissues, particularly human gastrointestinal tissue, observed as part of routine clinical practice among ALS patients prior to diagnosis of their motor symptoms [135]. It is unlikely that TDP-43 aggregates in the gut are pathogenic but they may well mirror a similar long preclinical accumulation in the brain. It is unclear if TDP-43 aggregates in the gut are relevant in the breakdown of the gut barrier, dysbiosis, and impairment of the gut–brain axis. 

## 9. Pathways to ALS from MNPLs

Key pathogenic elements of ALS are mislocalization and aggregation of the DNA/RNA-binding protein TDP-43 [136], cortical hyperexcitability [137,138], and neuroinflammation [139,140,141]. Figure 3 depicts how these may interact in response to MNPL neurotoxicity. TDP-43 is an essential protein involved in several DNA and RNA processing events, including gene transcription, pre-RNA splicing, RNA translation and degradation, and critical for normal neuronal health. Although it mainly localizes in the nucleus, TDP-43 also shuttles between the nucleus and cytoplasm for various physiologic functions. Under stress conditions, such as those induced by MNPLs, it is initially transiently recruited into membrane-less stress granules in the cytoplasm [142]. There is a mechanistic link between stress granule formation and development of pathological TPP-43 inclusions [143]. Even small changes in TDP-43 levels and its intracellular localization are highly predictive of neurodegeneration [144]. Loss of TDP-43 cryptic splicing repression occurs early in disease progression and is detectable pre-symptomatically [145]. TDP-43 CSF concentrations have been positively correlated with nanoplastics [146,147]. Inflammatory-mediated oxidative stress is a ubiquitous response to MNPL-induced inflammation, and even small doses are toxic when exposure is chronic [9]. 

MNPLs are known to induce neuronal excitotoxicity [148]. For example, polystyrene nanoplastics induce neuroexcitation in zebrafish larvae [149]. In a *C. elegans* model, weathered nanoplastics significantly reduced both glutamate and GABA transmitter function, but GABA to a greater extent [150]. Disrupted GABAergic circuits shift the balance between cortical excitation and inhibition (excitatory/inhibitory balance), a crucial aspect for normal brain function [151], and more specifically finely tuned motor function mediated by monosynaptic corticomotoneuronal projections [152]. 

Evidence from human electrophysiological studies and ALS animal models strongly supports cortical hyperexcitability as important in the pathogenesis of ALS, occurring prior to the onset of the clinical syndrome [138,153]. There is a clear link between cortical hyperexcitability and subsequent downstream degeneration and loss of anterior horn cells [154,155]. Cortical hyperexcitability is a combined consequence of increased excitatory (glutamatergic) inputs to the upper motor neuron, paralleled by decreased inhibition mediated through GABAergic interneurons. Evidence from human transcranial magnetic stimulation studies indicates decreased GABAergic inhibition plays the dominant role in cortical excitability [156]. GABA_A_ receptors in the motor cortex of patients with ALS show downregulated α1-subunit and upregulated β1-subunit mRNAs indicating altered receptor function [157]. The resulting imbalance in excitation/inhibition is shared by other neurodegenerative diseases [158,159], and may be an early sign leading to disease in later years [95]. Studies in *C. elegans* indicate that glutamate and especially GABA are significantly reduced after administration of weathered microplastics [150,160], resulting in net excitotoxicity.

## 10. Preventive and Therapeutic Measures

Plastic control strategies are being vigorously explored and are becoming effective for mitigating plastic pollution and its impact on the environment ([11] and https://www.nih.gov/news-events/news-releases/microplastics-algal-blooms-seafood-safety-are-public-health-concerns-addressed-new-oceans-human-health-centers 16 April 2024). Single-use products made of polymeric plastics (drinking bottles, straws, cutlery, coffee cups, and bags), are recognized as a significant source of plastic pollution, and there is increasing legislation preventing their use. However, some plastic materials, for example facemasks, prominently used during the coronavirus epidemic (COVID-19), are challenging to recycle. Using biodegradable plastics and changing individual behaviours has its own challenges, making it important to ensure that all aspects of microplastic issues, including their origins, types, effects, and fates, are widely publicized. Of concern is MNPL toxicity occurring during the neonatal period, setting the stage for later life neurodegeneration, as well as multigenerational epigenic influences. Our focus needs to be on strategies for their successful removal from air and aquatic ecosystems. These include coagulation, membrane bioreactor technology, rapid sand filtration, and adsorption, in addition to more innovative techniques such as electrocoagulation, photocatalytic degradation, electrochemical oxidation, and magnetic separation [11].

Ingestion is the prime source of plastic toxicity in humans, resulting in gut dysbiosis as an early step. Adopting a diet rich in fiber, whole grains, fruits, and vegetables can promote the growth of beneficial bacteria in the gut, along with avoiding processed foods. Probiotics help restore balance to the gut microbiota and because prebiotics are non-digestible fibers, they enhance beneficial bacteria in the gut [118]. More specifically, the prebiotic role of polyphenols influences gut microbiota in neurodegenerative disorders by modulating intracellular signaling pathways. Metabolites of polyphenols function directly as neurotransmitters by crossing the blood–brain barrier [161]. Another approach is use of fecal microbial transplantation, which aims to restore eubiosis, rebalance gut microbiota, and reestablish immunological tolerance. A recent double-blind, controlled, multicenter study using fecal microbial transplantation in ALS was found to modulate neuroinflammation, modifying disease activity and progression [162].

The glymphatic system and the meningeal lymphatic vessels provide a pathway for transport of solutes and clearance of toxic material from the brain [163]. Impairment of this system has been implicated in several neurodegenerations in association with pollutants [164]. Of specific relevance to ALS, this is applicable to TDP-43 and glutamate, both major elements in disease pathogenesis. The glymphatic system clears metabolic waste in the brain by exchanging the interstitial fluid (ISF) surrounding neurons with ‘clean’ cerebrospinal fluid (CSF) [165]. Clearance includes removal of toxic proteinaceous aggregates including TDP-43 [166], and recent studies show that neurons actively drive glymphatic clearance of waste [165]. There is an important role for glymphatic dysfunction in ALS pathology, which correlates with sleep disturbances in early-stage ALS [167]. MNPLs may impair glymphatic functioning directly or indirectly and there are strategies directed at improving glymphatic clearance, which may also reduce MNPL-induced neurotoxicity [168].

## 11. Conclusions

Neurotoxicity and other health hazards related to the exponential growth of MNPLs are new. MNPL-related human health hazards are becoming recognized but information has been largely derived from non-human species. MNPLs, especially when weathered, can breach the BBB or access the brain through nose-to-brain translocation. However, studies in ALS specifically related to MNPL toxicity have not been reported and there have been no neuropathological studies, or reports of specific biomarkers related to microplastics or nanoplastics in ALS. The exposome, referred to earlier in this review, integrates external exposures arising from the outside milieu, their biological fingerprints in biofluids, and susceptibility factors that modulate biological responses to the environment. New advances in exposomics in neurodegeneration will enhance our knowledge of the toxicological effects of MNPLs in ALS [163]. The resulting neurotoxicity is reflected in an inflammatory response with intracytoplasmic stress granule formation and subsequent formation of TDP-43 aggregates, a hallmark of ALS. Neurotransmitter anomalies involving glutamate and GABA, primarily shown in non-human species, result in altered excitation/inhibition balance, favoring loss of inhibition. This induces cortical hyperexcitability, another important pathogenic mechanism in ALS. In humans, ingestion is the commonest route of exposure to MNPLs, and causes acute and chronic gut inflammation with breakdown of the gut barrier. Resulting dysbiosis of the gut microbiome with disruption of the gut–brain axis is implicated in neurodegenerative diseases, including ALS. There is significant potential for exposure to MNPLs during the neonatal and early childhood periods through breast milk, milk substitutes, and toys. The low-grade chronic neurotoxicity this induces may set the stage for later life disease including neurodegeneration. Exposure to MNPLs should be considered a risk factor for ALS and other neurodegenerative diseases.

## Figures and Tables

**Figure 1 brainsci-14-00471-f001:**
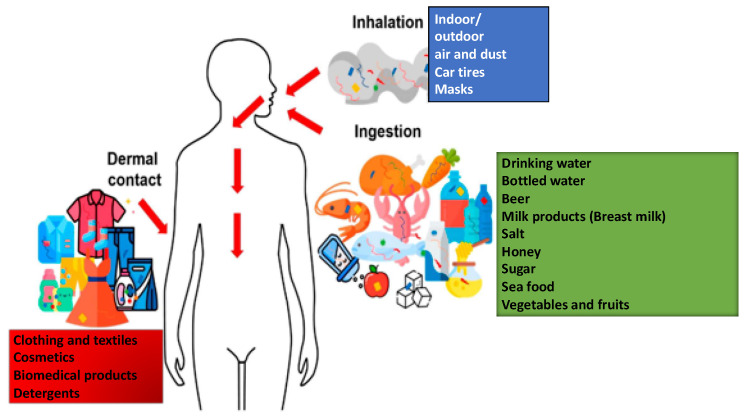
Human exposure to MNPLs. Human exposure to MNPLs through inhalation of indoor and outdoor air and dermal contact happens through clothing and self-applied cosmetics, but also biomedical products. In humans, exposure mainly occurs through ingestion of contaminated food and water. Neonates and infants are exposed to toxic plastics through breast milk and milk substitutes, but also by chewing on plastic-polluted toys.

**Figure 2 brainsci-14-00471-f002:**
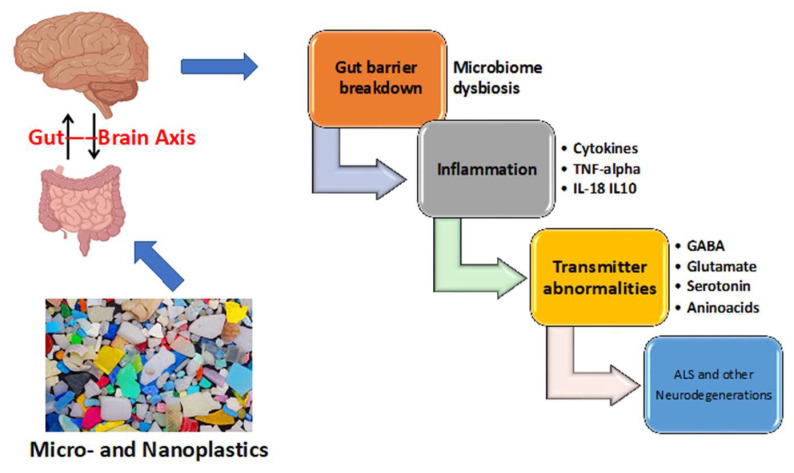
MNPLs and the gut–brain axis: Ingestion is the prime entry route of MNPLs in humans. These pollutants break the gut barrier, cause microbiome dysbiosis and then induce an inflammatory reaction which may be chronic. Impairment of the gut–brain axis is implicated in the pathogenesis of ALS.

**Figure 3 brainsci-14-00471-f003:**
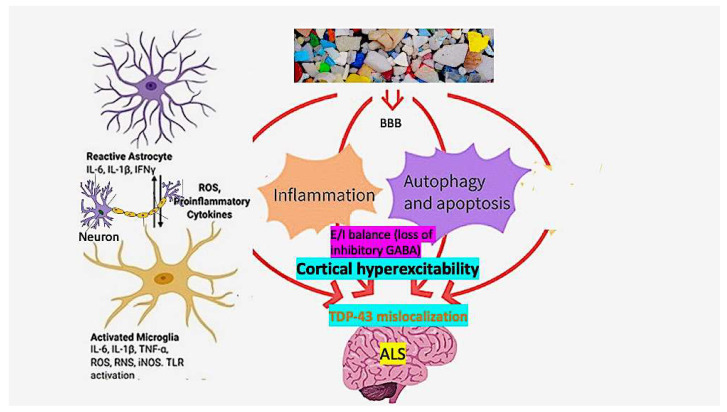
A route from micro/nanoplastic pollution to ALS: In non-human species, MNPLs induce a neuroinflammatory cascade involving activated microglia and astrocytes with mediation of key pro-inflammatory cytokines (IL-1β, IL-6, and TNFα) and oxidative stress to induce stress granules, with mislocalization of TDP-43 and aggregate formation. Neurotransmitter anomalies develop in favor of GABAergic inhibition and impaired excitatory/inhibitory (E/I) balance with a net increase in cortical excitability. The process is likely chronic and may commence in early years of life. These events, especially neuroinflammation, are common to all neurodegenerations.

**Table 1 brainsci-14-00471-t001:** Sources of MNPLs.

Primary Sources:
Breakdown of larger plastic items such as packaging materials, bottles, and synthetic textiles.
Microbeads used in personal care products.
Emission of plastic particles during manufacturing processes.
2. Secondary Sources:
Effects due to weathering, sunlight, and wave action.
Tire wear with release of microplastic particles.
Deterioration of road markings containing microplastics.
3. Urban Runoff:
Flushing of microplastics into rivers and oceans.
Incomplete removal of microplastics during wastewater treatment.
4. Atmospheric Deposition:
Microplastics transported by the wind and deposited from the atmosphere.
5. Agricultural Runoff:
Plastic mulching with release of microplastics into the soil.
Plastic particles from irrigation systems.
6. Natural Sources:
Natural weathering can release microplastics from rocks and sediments.

## Data Availability

No new data were created or analyzed in the preparation of this review. Data sharing is not applicable to this article.

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
