# Peer review of "Nanoplastics and Neurodegeneration in ALS"

_brainsci, 2024, doi:10.3390/brainsci14050471_

Round 1
Reviewer 1 Report
Comments and Suggestions for Authors
I appreciate the opportunity to review the manuscript entitled "Nanoplastics and neurodegeneration in ALS". The area discussed in this manuscript is complex, because ALS's pathophysiological mechanisms are also extremely complex, involving an individual or familial basis of genetic predisposition and the direct and indirect impact of environmental factors. However, the authors have produced an interesting content summarizing all the main topics currently discussed about the potential impact of nanoplastics and microplastics in the pathogenesis of neurodegenerative processes. Figures 1 and 2 are important key aspects of the manuscript: their key messages are properly transmitted to the reader. Two points could be assed by the authors at this time for review:
1. I have only a minor suggestion at this point that authors should include a paragraph (or phrase) in their text describing that this narrative review manuscript and its discussion and conclusion were based on the analysis of several distinct sources from the literature and that there were no neuropathological studies from autopsies (or necropsies), for example, disclosing specific findings or biomarkers related to microplastics or nanoplastics.
2. Other interesting topic which could be discussed briefly is if the authors have any idea of potential role of glymphatic system in the neuropathogenesis of ALS in the context of nanoplastics.
Author Response
We thank the reviewer for the helpful comments, these are appreciated.
I have only a minor suggestion at this point that authors should include a paragraph (or phrase) in their text describing that this narrative review manuscript and its discussion and conclusion were based on the analysis of several distinct sources from the literature and that there were no neuropathological studies from autopsies (or necropsies), for example, disclosing specific findings or biomarkers related to microplastics or nanoplastics.
We have added several sentences in the conclusion that clearly states that studies in humans and in particular ALS have not been explored.
Other interesting topic which could be discussed briefly is if the authors have any idea of potential role of glymphatic system in the neuropathogenesis of ALS in the context of nanoplastics.
We agree this is a very interesting issue and have tackled some relevant points under the section on "prevention and therapeutics"
Reviewer 2 Report
Comments and Suggestions for Authors
In the article entitled "Nanoplastics and neurodegeneration in ALS", the authors review how plastics may be a cause of ALS.
They state that nanoplastics could be a cause of mislocalization of TDP-43, a pathological hallmark of the disease.
The article is interesting as it discusses how pollution and exposure to nanoplastics could be a cause of neurodegenerative diseases such as ALS.
It would be interesting if the authors would mention the studies that have been carried out, if any, either in vitro and/or in vivo to study this effect.
Figure 3 is one approach to the mechanism; however, many neurodegenerative diseases have neuroinflammation in common.
Link how nanoplastics can generate mislocalization of TDP-43.
Describe whether other genes characteristic of the disease could be affected by exposure to nanoplastics.
Author Response
We thank you for your helpful and constructive comments.
It would be interesting if the authors would mention the studies that have been carried out, if any, either in vitro and/or in vivo to study this effect.
As mentioned in several of the segments of the paper, there is very limited direct human studies and none in ALS. There are a few studies mentioned in the text related to IPSc cell lines. Also described are some of the difficulties related to determining toxicity due to MNPLs.
Figure 3 is one approach to the mechanism; however, many neurodegenerative diseases have neuroinflammation in common.
We have added a note to figure 3 to indicate the commonality of inflammation amongst neurodegenerations.
Link how nanoplastics can generate mislocalization of TDP-43.
There is no clear information on this - but there is recent evidence described in the text now indicating that TPD-43 aggregates occur in the gut long predating the origin of ALS motor symptoms.
Describe whether other genes characteristic of the disease could be affected by exposure to nanoplastics.
Again there is no solid information regarding the relationship if any between the numerous identified risk genes in ALS and the few causative genes and plastic toxicity. Maybe more plausible is the effect on the neonatal epigenome that has been referred to.
Round 2
Reviewer 2 Report
Comments and Suggestions for Authors
Thanks to the authors for their replies. Based on the suggested changes, the manuscript improved considerably.